# A 320 μW Multi-Band Receiver with N-Path Switched-Capacitor Networks

Lei Lei [1], Fang Han [1], Zicheng Liu [2], Quanwen Qi [2], Xinghua Wang [1] and Weijiang Wang [1,*]

1  Beijing Institute of Technology, Beijing 100081, China
2  Beijing Institute of Technology Chongqing Center for Microelectronics and Microsystems, Chongqing 400030, China
*  Correspondence: wangweijiang@bit.edu.cn

**Abstract:** This paper presents a multi-band ultra-low power (ULP) receiver with N-Path Switched-Capacitor (NPSC) networks in 90 nm CMOS process. The NPSC is integrated into the feedback loop of the low noise amplifier (LNA) to flexibly provide narrowband input matching at multiple sub-GHz Industrial, Scientific, and Medical (ISM) bands by adjusting the switching frequency. Moreover, the LNA with an NPSC network is utilized to suppress the out-of-band signal at the input and output of the LNA, simultaneously. In order to achieve an ultra-low power consumption, a sub-threshold LNA and four passive NPSC mixers are implemented in this receiver. The ULP receiver achieves a measured gain of $40 \pm 2$ dB in ISM bands (430/860/915/960 MHz). The measured noise figure and out-of-band IIP3 are $10 \pm 0.5$ dB and $-0.3 \pm 2$ dBm, respectively. The ULP receiver chip consumes 320 μW at 0.4 V power supply and occupies a chip area of 0.31 mm$^2$.

**Keywords:** CMOS; N-Path Switched-Capacitor; passive mixer; receiver; ultra-low power

## 1. Introduction

The Internet of Things (IoT) has had a revolutionary impact on human life and has become a topic of great concern in research and industry [1–5]. Ultra-low power (ULP) receivers are the cornerstone of various IoT applications such as the industrial internet, intelligent buildings, medical treatment, and consumer electronics. Generally, IoT devices are powered by batteries. Due to the limited lifespan of batteries, there is a strong demand for self-powered devices or alternative energy sources to continuously power IoT devices. As the IC industry keeps up with the development of sensors, ULP wireless communication and batteryless operation will become two key enabling technologies. Some current research directly supply power through sub 0.5 V energy sources [6–9]. In particular, the design of a multi-band ULP receiver with direct power supply below 0.5 V has attracted great interest [10–14].

Circuit imperfections, especially in power-consuming blocks, poses a real challenge to ULP radio design. The power consumption of the receiver mainly comes from two parts. On the one hand, the signal amplification in the RF stage consumes high power. On the other hand, high linearity and sharp roll-off filters cause high power. Most of the existing ULP receivers are optimized based on the above two aspects. In order to reduce the power of the RF gain stage, Refs. [15–19] chose to eliminate the RF amplifier stage and used the mixer as the first stage of the receiver. However, the mixer-first receiver suffers from a higher noise figure, which degrades the sensitivity of the receiver. Therefore, it is also necessary to pay attention to the consideration of the trade-off between power consumption and noise when dealing with the RF gain stage. Refs. [20,21] use passive RF gain instead of active RF gain, but the single-stage passive gain is insufficient. An additional off-chip transformer is needed in [20] to complete the input matching, which reduces the integration of the receiver.

The interference suppression of the filter requires high power to synthesize the high-Q filter response. Previous studies have achieved high-Q filter response through multi-stage active filter. Recently, N-path filters have received widespread attention because of their ability to provide high-Q filtering around digitally programmable switching frequencies [22–26]. In addition to being a filter, the N-Path Switched-Capacitor (NPSC) network also can be used for down-conversion in the mixer-first structure [27–31]. Figure 1 shows a simple model of an NPSC network, where $R_{sw}$ is the on-resistance of the switches. The switches are periodically turned on without overlapping, and the capacitors on each path periodically store charges. Based on the time-variant nature combined with the memory effect of the capacitors, the NPSC network realizes the band-pass filtering whose center frequency is accurately provided by the switching frequency. The switches also down-convert the RF signal, not only outputting the baseband voltage at the capacitor, but also filtering at the RF input. Recent papers present complicated mathematical analysis of the NPSC network by using the linear periodically time-variant (LPTV) theory [32–36]. Reference [32] obtained the equivalent RLC model of the N-path filter by deriving the transfer function of the N-path filter. Based on the characteristics that the N-path filter is equivalent to an RLC network, Refs. [37,38] applied the N-path filter to achieve impedance matching. Reference [37] combined N-path filter and amplifier to obtain a gain boosted low noise amplifier (LNA). Ref. [38] designed an LNA with translational positive feedback from baseband to RF through a four-phase switching mixer.

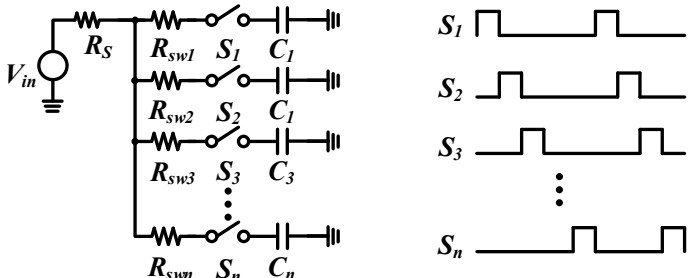

**Figure 1.** NPSC network with multiphase non-overlapped clocks.

In this work, a ULP receiver is realized by utilizing various functions of the NPSC networks. In order to overcome the large power consumption of the RF stage, a single-stage amplifier biased in the sub-threshold region of the RF amplifier is applied. A low power supply voltage employs to satisfy the direct power supply requirements of environmental energy. In addition, the NPSC network is embedded in the feedback loop of the RF amplifier to provide sufficient gain while reducing the power consumption. Furthermore, the passive NPSC network is used as a mixer to achieve signal down-conversion. This brief is organized as follows: Section 2 presents the overall architecture of the receiver and the circuit implementation, and analyzes the design of the ultra-low power consumption in the LNA, the mixer, and the divider. Section 3 reports the experimental results of the receiver fabricated in 90 nm CMOS and discusses the results. Finally, Section 4 summarizes this paper.

## 2. ULP Receiver with NPSC Networks

As illustrated in Figure 2, the proposed receiver comprise an LNA, four passive mixers, four IF amplifiers, and a divider. The LNA amplifies the RF signals and provides input impedance that matches with the antenna. Moreover, the LNA possesses the capability to suppress the out-of-band signal at the input and output of the RF amplifier, respectively, which achieves the same filtering function as a band-pass filter. Subsequently, the LNA precedes the mixers driven by four-phase non-overlapped local oscillator signals with 25% duty cycle, which is generated by the divider. The RF signal flows into one of the four mixers without overlapping since only one of the four mixers is turned on in the down-conversion operation. As a consequence, the RF current flows into four mixers

on average in one clock cycle. Finally, the IF amplifier follows the mixers to amplify the IF signal.

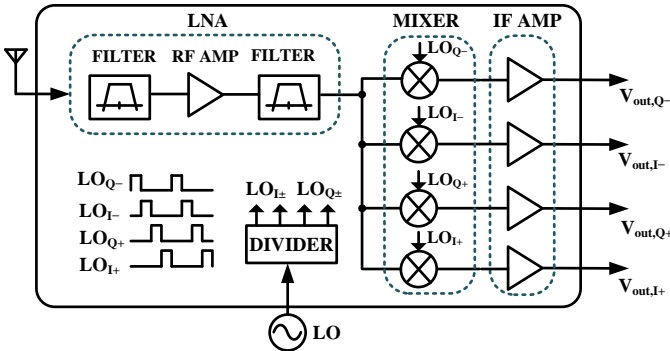

**Figure 2.** Block diagram of the ULP receiver with NPSC networks.

The schematic of the proposed receiver is shown in Figure 3. The LNA acting as a transconductance device to output RF current incorporates an inverter amplifier biased in the sub-threshold region and an NPSC network embedded in the feedback path. In particular, the NPSC network provides frequency-selectable filtering for the LNA. Moreover, the input impedance exhibits frequency selection property, which can be matched to 50 Ω at different ISM bands by changing the frequency of the switches. The switches in the NPSC network are controlled by four-phase non-overlapped local oscillator signals with 25% duty cycle, which is the same as the LO signals indicated in Figure 2. The second stage is composed of four passive NPSC mixers. Capacitor $C_c$ is adopted for AC coupling between the two stages. After the IF current is "sharped" by the low-pass load for the mixer, the IF signal is amplified respectively by four IF amplifiers with the same structure as the LNA.

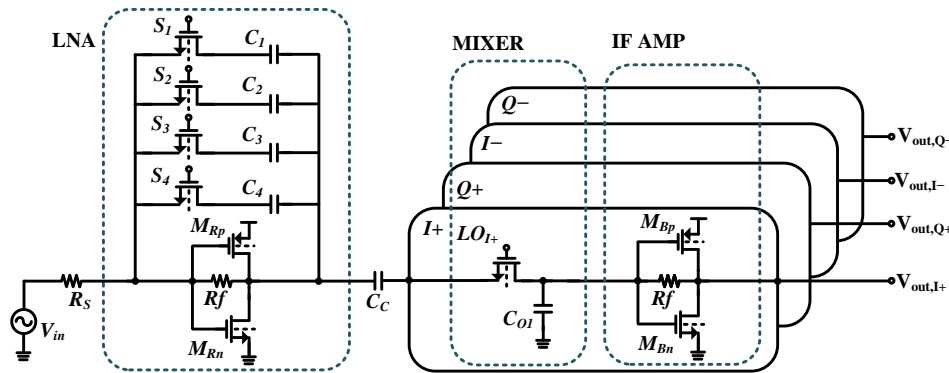

**Figure 3.** Schematic of the ULP receiver with NPSC networks.

### 2.1. LNA with an NPSC Network

The proposed LNA with an NPSC network is depicted in Figure 4a. A sub 0.5 V power supply voltage is applied to satisfy the design specification of the system powered by environmental energy while to save power consumption. The current signal amplified by the RF amplifier forms a voltage drop at the input through the feedback network. Since the feedback network senses the output current and returns a voltage, its feedback factor has the dimension of resistance. In the frequency domain, the band-pass characteristic at the switching frequency on the feedback loop is reflected in the input impedance. Since the change of the switching frequency corresponds to the change of the resonance frequency, the input matching network possess a variable frequency characteristic. The resistance and band-pass characteristics of the NPSC network are quantitatively analyzed in the previous research [24], and an equivalent RLC model is established. The resistance and capacitance

are determined by the duty cycle and the switch on resistance. The inductance is closely related to the switching frequency and resonant out with capacitance at switching frequency.

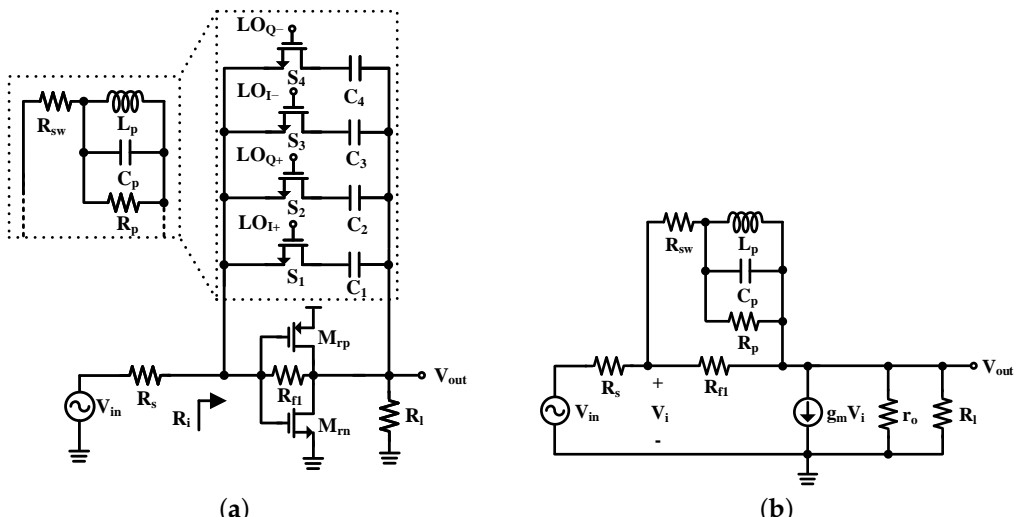

**Figure 4.** (**a**) Proposed LNA with an NPSC network and its equivalent RLC circuit. (**b**) Small-signal equivalent circuit.

The derived RLC network shown in Figure 4a is used to analyze the circuit characteristics of the NPSC network in the feedback loop in this work. The small-signal equivalent circuit of the proposed LNA with an NPSC network is shown in Figure 4b. $R_{sw}$ is the mixer switch on-resistance. $R_{f1}$ as a feedback resistance of the RF amplifier forces the output bias voltage to a known value, which is much larger than the on-resistance of the switch. For simple analysis, $R_{f1}$ is ignored. $R_l$ is the input impedance looking into the mixer, which is demonstrated in section B. In addition, the $g_m$ and $r_o$ of the NMOS and PMOS are combined in the small signal model, $g_m = g_{mn} + g_{mp}$ and $r_o = r_{on} + r_{op}$. The input impedance in and out of the band is derived as follows, where $r_o$ represents the MOS output resistance.

$$R_i|_{f=f_s} = \frac{R_p + R_{sw} + r_o \parallel R_l}{1 + g_m(r_o \parallel R_l)} = R_s \tag{1}$$

$$R_i|_{f=f_s \pm \Delta f_s} = \frac{R_{sw} + r_o \parallel R_l}{1 + g_m(r_o \parallel R_l)} \approx \left( \frac{R_{sw}}{r_o \parallel R_l} + 1 \right) \frac{1}{g_m} \tag{2}$$

For simple analysis, $g_m(r_o \parallel R_l) \gg 1$ is assumed. It can be obtained from (1) that the influencing factors of input impedance include $g_m$, $R_{sw}$, $r_o$, and $R_l$. These greatly increase the freedom of the design. In order to achieve better out-of-band rejection, the out-of-band input impedance value needs to be as small as possible. In (2), the out-of-band input impedance is inversely proportional to the transconductance. Transconductance biased in the subthreshold region is $I_d q / nkT$, which has a linear relationship with drain current. Therefore, a larger drain current is required to improve the out-of-band suppression. The subthreshold drain current is determined by the aspect ratio and the gate-source voltage. Unlike working in the strong inversion region, the transconductance and voltage headroom are not restricted by the lower bound of the threshold voltage in the subthreshold region. For this reason, the gate voltage with half of the power supply voltage is designed to maximize the swing. Furthermore, a large aspect ratio is significant to increase the out-of-band suppression. However, the LNA has relatively high parasitic capacitance as the size of the device increases. Thus, the performance of the input matching and the out-of-band rejection is a trade-off, since excessive parasitic capacitance severely affects the input matching.

In the same way as (1) and (2), by analyzing the small-signal equivalent circuit of the proposed LNA with NPSC network shown in Figure 4b, the in-band and out-of-band gains of the designed LNA are shown in (3) and (4), respectively.

$$A_v|_{f=f_s} = \frac{(r_o \parallel R_l)[1 - g_m(R_p + R_{sw})]}{R_s[1 + g_m(r_o \parallel R_l)]} \tag{3}$$

$$A_v|_{f=f_s \pm \Delta f_s} = \frac{(r_o \parallel R_l)(1 - g_m R_{sw})}{R_{sw} + r_o \parallel R_l} \tag{4}$$

With the increase of $g_m$, the gain boost in the band is much greater than the gain out of the band. The greater the value of $g_m$, the greater the difference between the in-band and out-of-band gain. Therefore, by enhancing $g_m$, the out-of-band suppression is considerably developed at the output of the LNA.

To exemplify, the circuit shown in Figure 4a is simulated. The above circuit is simulated by cadence virtuoso IC61 software. The components in the circuit use the models provided by the TSMC 90 nm process library. The simulated power supply voltage is 0.4 V. From Figure 5, the simulation results for four ISM bands of 430 MHz, 860 MHz, 915 MHz, and 960 MHz are shown. By changing the transistor width of $M_1$ and $M_2$, the value of $g_m$ is changed, while adjusting the value of the feedback loop capacitance to satisfy impedance matching. The parameters are: $R_{sw} = 20\ \Omega$, $R_{f1} = 80\ k\Omega$, $R_l = 10\ k\Omega$, and $R_s = 50\ \Omega$. With the increase of $W$, the rejection of out-of-band input and the rejection of out-of-band gain are significantly improved. However, considering the limitation of power consumption, the value of $W$ should not be too large.

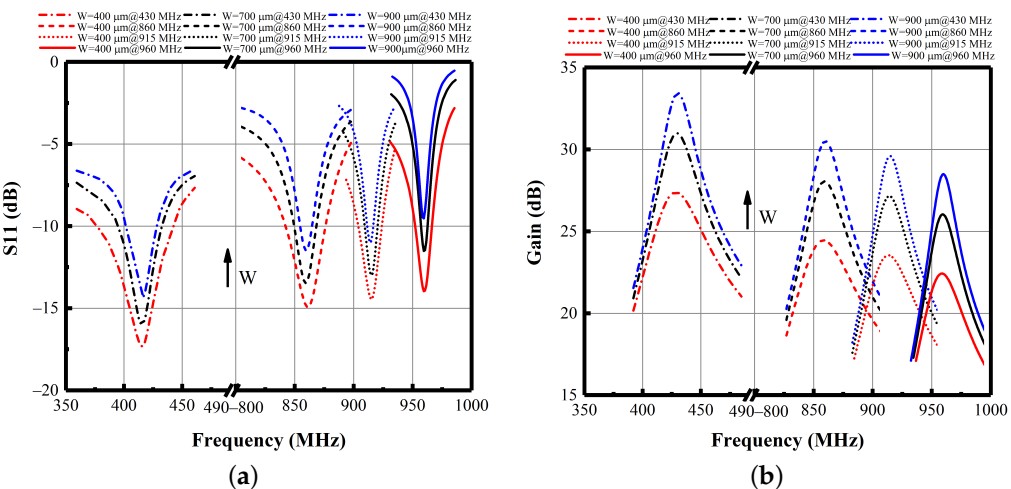

**Figure 5.** Simulated (**a**) S11 and (**b**) Gain of the LNA for four ISM bands of 430 MHz, 860 MHz, 915 MHz, and 960 MHz.

The proposed LNA utilizes sub-threshold biased devices, allowing supply voltages below 0.5 V while saving power consumption. The IF amplifier is designed with the same structure as the LNA to provide sufficient gain for the IF signal. A supply voltage below 0.5 V results in the severe headroom limitation, and the designed amplifier employing NMOS and PMOS for amplification satisfies the headroom requirement and doubles the transconductance and gain for the same bias current.

### 2.2. NPSC Mixer

The NPSC mixer is shown in Figure 6a, which is driven by periodic four-phase non-overlapped local oscillator signals with 25% duty cycle. The LNA in the previous stage views as an RF current source to provide RF current and a output impedance in series. Thus, considering the NPSC mixer as a passive mixer current conveyor shifts the RF current. The RF current flows through the current-driven passive mixers, and the down-converted

current signal passes through the baseband load to obtain the output voltage. Due to the similarity of the four IQ paths, the $I+$ path is analyzed below.

As shown in Figure 2, $LO_{I+}$ is the control signal of the mixer, which is a square wave toggling between 0 and 1. Thus, the switching operation is equivalent to the input signal multiplied by $LO_{I+}$. Moreover, $z_{BB}(t)$ is the impulse response of the load of the mixer.

$$V_{BB,I+}(t) = [LO_{I+}(t)i_{RF}(t)] * z_{BB}(t) \tag{5}$$

The output voltage is affected by the down-converted current and the frequency response of the load. As shown in Figure 6b, the RF signal is down-converted to the baseband by the NPSC mixer in the first place. Assuming that the load is a low-pass impedance such as RC series, the output shown in (5) is affected by the frequency response of the load, so the output voltage indicates low-pass characteristics. Intuitively, the RF signal up-converted by the mixer exhibits band-pass characteristic. Therefore, it can be seen that the mixer exhibits filtering behavior due to the frequency response of the load. Owning to this transparency of the passive mixer, the input impedance of the next stage is equivalently transferred to the input port of the mixer. In this design, the output resistance $R_l$ of the LNA stage has the same response as the input impedance of the IF amplifier.

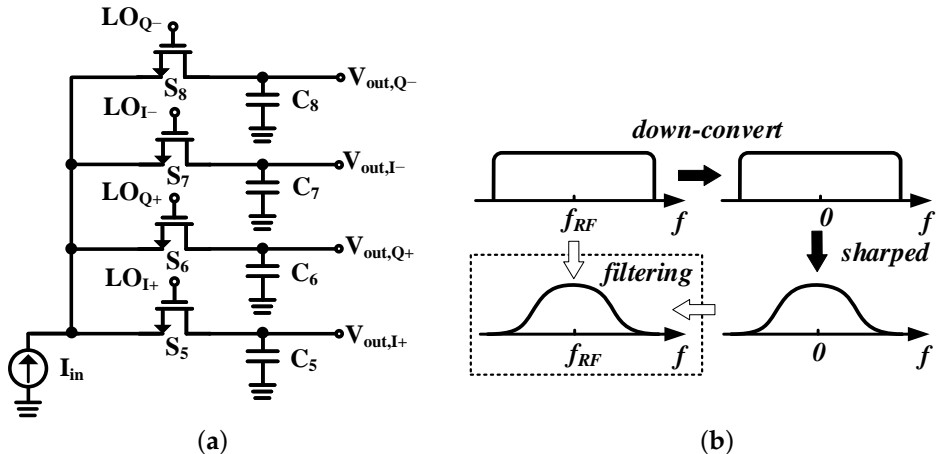

| (a) | (b) |

**Figure 6.** (**a**) Schematic of the NPSC mixer. (**b**) Spectra of the NPSC mixer at input and output.

### 2.3. Divider

On-chip logic gates are employed to generate four non-overlapping sharp local oscillator signals with rise and fall times much lower than the clock period. The divider in Figure 7 consists of four pull-up devices and two regeneration loops composed of NOR gates. A four-phase 25% local oscillator signal is generated from a square wave with a 50% duty cycle. First, the clock is applied as a gating signal, and the four outputs are only valid when the clock is high. Second, channels I and Q use the clock and the reverse of the clock as the gate control signals, which are valid at the high level and low level of the clock. Thus, every two cycles of the clock signal is divided into four local oscillator signals with 25% duty cycle.

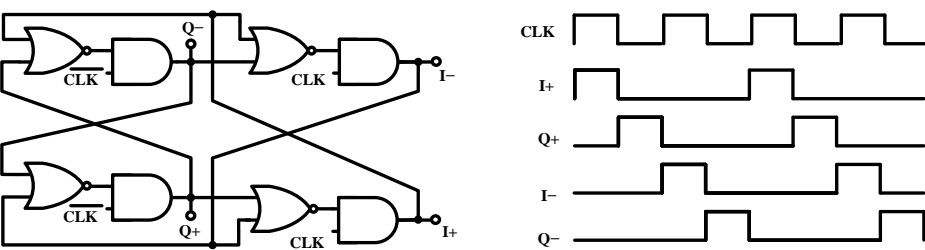

**Figure 7.** Generation of 25% duty-cycle quadrature clocks.

### 3. Measurement Results

The receiver with NPSC networks is fabricated in a 90 nm CMOS technology and the die photo is shown in Figure 8a. The receiver chip occupies an area of $860 \times 360$ μm$^2$ excluding pads. The chip is assembled on a printed circuit board (PCB) and measured to validate the good performance. Figure 8b shows the receiver chip on the PCB assembly. The I/Q paths are transformed from the four output signals through the off-chip balance. Four ISM frequency bands of 430 MHz, 860 MHz, 915 MHz, and 960 MHz, which are mainly used for IoT applications, were tested.

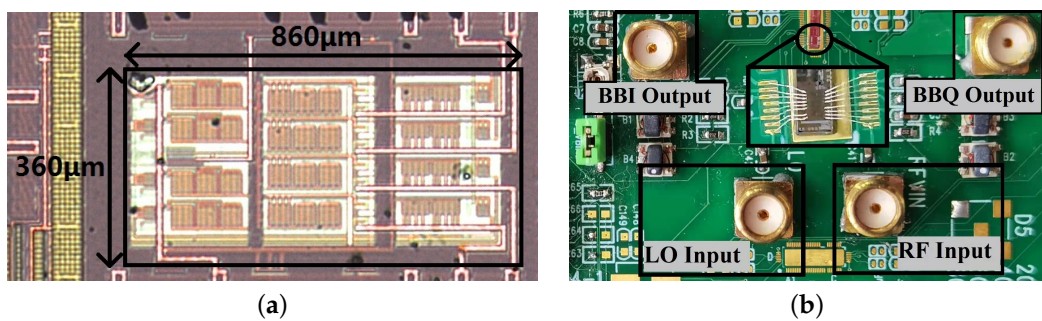

|     |     |
| :-: | :-: |
| (**a**) | (**b**) |

**Figure 8.** (**a**) Chip micrograph. (**b**) Assembled PCB for testing.

The testing setup is shown in Figure 9. A DC power supply is connected to the chip to provide DC power. A signal generator is applied to the clock input. The RF input is provided by the vector network analyzer and the output is connected to the vector network analyzer. In addition, the reference clock of the signal generator is connected to the vector network analyzer so that the clocks of the two instruments are synchronized.

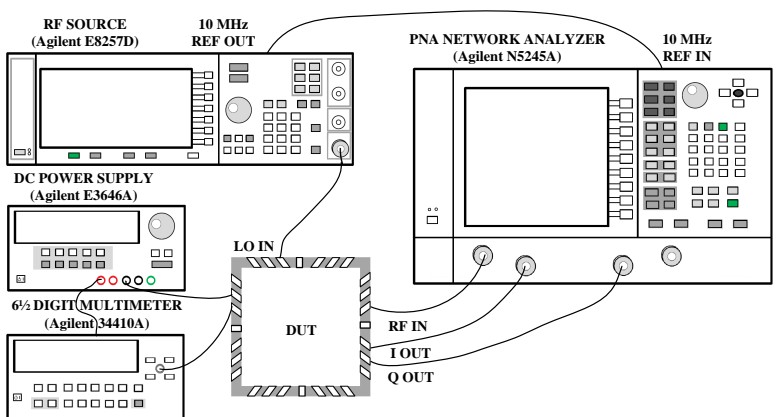

**Figure 9.** ULP receiver system testing setup.

In Figure 9, a $6\frac{1}{2}$ digit multimeter is connected in series between the DC power supply and the chip to measure the power consumption of the chip. The proposed receiver draws a current of 0.8 mA from a supply voltage of 0.4 V. The power consumed by each block is measured individually by controlling the block with an on-chip enable control switch. The NMOS is used as a switch to control the on-state of the power supply voltage of each block. When measuring a certain block, other blocks are in a short circuit state. We turn on each block in turn to obtain the power consumed by each block during power consumption measurement. The power breakdown of the implemented receiver is shown in Table 1. The power consumption of the LNA, IF amplifier, and divider are 134 μW, 134 μW, and 52 μW, respectively. Thanks to the sub-threshold-biased devices and the passive NPSC networks design, the supply voltage could decrease to 0.4 V reliably.

**Table 1.** Power breakdown of the implemented receiver.

| Block | Power Consumption | Power Contribution |
|---|---|---|
| LNA | 134 µW | 42% |
| IF amplifier | 134 µW | 42% |
| Divider | 52 µW | 16% |
| Total | 320 µW | 100% |

The measured results of four ISM frequency bands of 430 MHz, 860 MHz, 915 MHz, and 960 MHz are shown in Figure 10. The proposed receiver is able to operate at any frequency of the ISM frequency band by selecting the corresponding input signal and the local oscillator signal. For example, the measurement result of 860 MHz is obtained by selecting the input signal as 860 MHz and the local oscillator signal as 850 MHz. When measuring the frequency of 915 MHz, we change the input signal to 915 MHz and the local oscillator signal to 905 MHz. Additionally, the receiver conversion gain is measured at an IF frequency of 10 MHz. In order to visually display the gain of each frequency band, the abscissa of the receiver conversion gain is selected as the radio frequency.

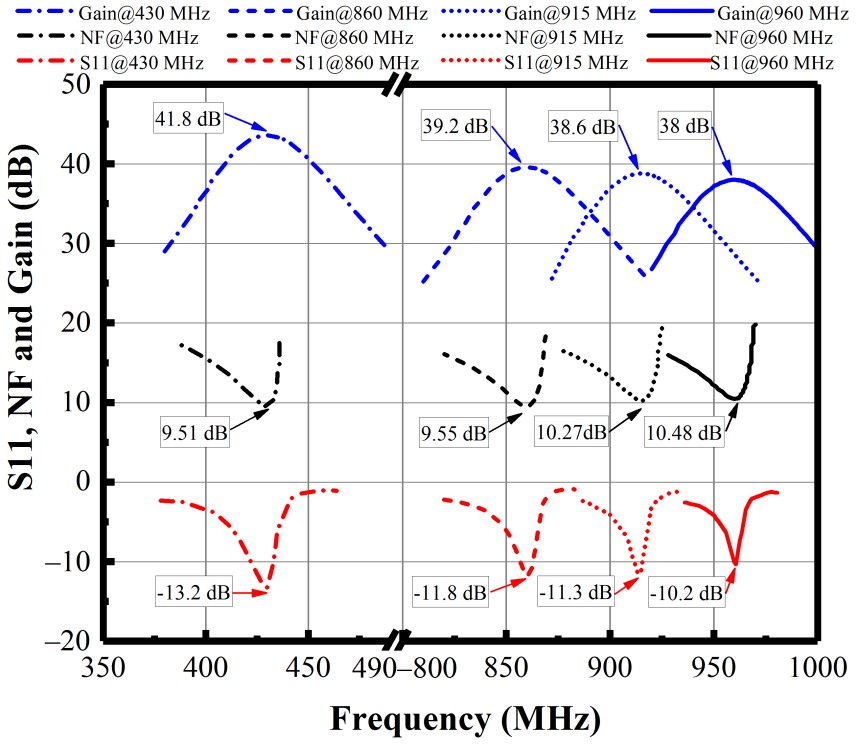

**Figure 10.** Measurement results of S11, NF and gain.

The S11 and the gain of the receiver shows the matching of signals of interest and also the matching of out-of-band signals. We observe better than −10 dB of S11 from the test bands. The measured noise figure of the receiver is $10 \pm 0.5$ dB, while the measured conversion gain is $40 \pm 2$ dB. We notice that, as the frequency increases, the gain decreases while the noise increases. This is because the parasitic effects become more pronounced as the frequency increases. Further, the leakage of the local oscillator has a greater impact at high frequencies. Consequently, the performance of the receiver is degraded at higher frequencies. Nonetheless, all measured performances are acceptable.

Measurements for the linearity performance are shown in Figure 11. It is measured by applying two tones of equal power at fs + Δf and fs + 2Δf + 1 MHz. The difference between the power of the third-order term generated by the two out-of-band signals and the power of the in-band signal is measured to indicate the out-of-band rejection of

the receiver. The out-of-band third-order intermodulation point can exhibit out-of-band rejection without indicating input power. Figure 11a shows that at a frequency offset of 100 MHz, the resulting OB-IIP3 is +1.72 dBm. The fundamental signal is measured at 860 MHz RF input and 850 MHz local oscillator frequency. Figure 11b shows the measurement results of OB-IIP3 with the offset frequency swept to 100 MHz. Figure 12b shows the measurement results of OB-IIP3 in other frequency bands. The measured OB-IIP3 in the ISM frequency band is $-0.3 \pm 2$ dBm.

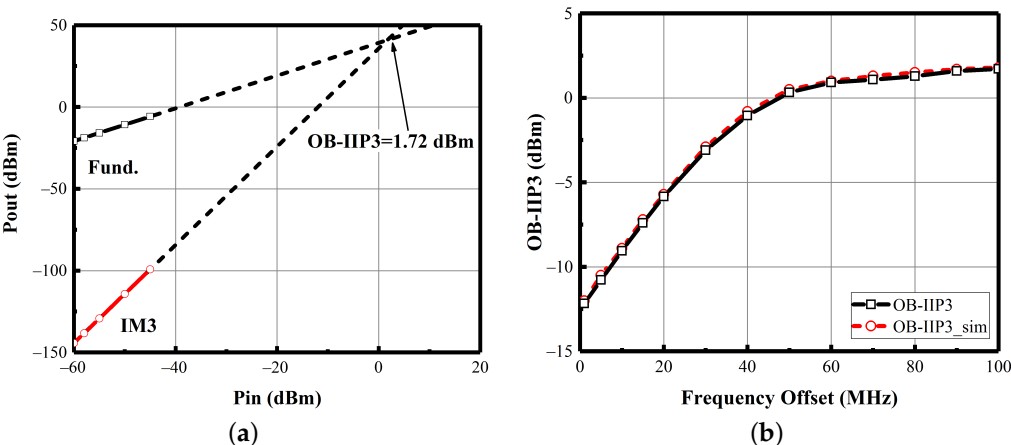

**Figure 11.** OB-IIP3 at 860 MHz (**a**) $\Delta f$ = 100 MHz. (**b**) $\Delta f$ = 1–100 MHz.

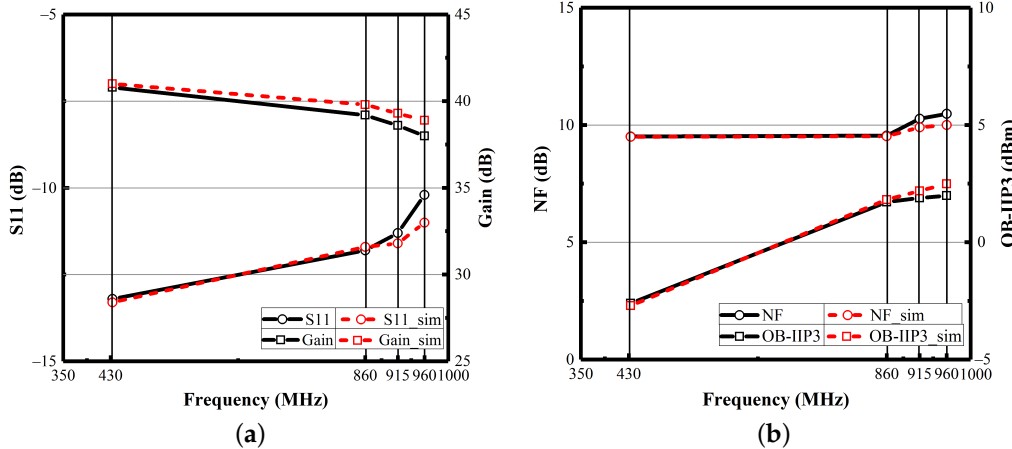

**Figure 12.** Simulated and measured performance (**a**) S11 and Gain. (**b**) NF and OB-IIP3.

Figure 12 shows the simulated and measured results. It can be seen that the simulation results are generally consistent with the measured results. In addition, it can also be used to roughly estimate the performance of the receiver in the entire 430–960 MHz frequency band.

Table 2 summarizes and compares the measurement performance of the implemented ULP receiver with the previous state-of-the-art work. It is found that the implemented receiver with NPSC networks has the lowest power consumption especially operate from a sub-0.5 V power supply voltage allowing direct powering from various energy harvesting sources. On the other hand, the proposed receiver is flexibility in the operating frequency. Compared with the wideband receiver and the receiver with a single working frequency, this receiver achieves the good performance in any narrowband channel in the ISM bands.

**Table 2.** Performance summary and benchmark with the state-of-the art.

| Reference | [1] | [38] | [39] | [40] | [41] | This Work |
|---|---|---|---|---|---|---|
| Tech./nm | 65 | 28 | 28 | 180 | 22 | 90 |
| Freq./MHz | 2400 | 2200–2400 | 850–2550 | 780/868/915 | 2400 | 430/860/915/960 |
| Power/μW | 860 | 580 | 530–970 | 1420 | 330 | 320 |
| VDD/V | 1 | 1 | 0.8 | 1 | 0.55 | 0.4 |
| Gain/dB | 57.8 | 19–42 | 55 | 45.9 | 32.3 | 40 ± 2 |
| NF/dB | 15.7 | 11.6 | 13.6 | 8.5 | 9.4 | 10 ± 0.5 |
| OB-IIP3/dBm | −13.4 | 3.3 | −7.5 | −33.5 [1] | −8 | −0.3 ± 2 |
| Area/mm$^2$ | 0.45 | – | 0.17 | 1.41 | 0.15 | 0.31 |
| Sensitivity/dBm | −88.3 | −96.4 | −90.4 | −102 | −89.6 | −94 ± 0.5 |
| FoM/dB$^2$ | 86.6 | – | 76.4–81.6 | 82.5 | 72.5 | 71 ± 0.5 |

[1] In-Band IIP3. dB re: 1 nW$^3$ mm$^2$.

Based on the SNR requirement of the ZigBee standard (7 dB SNR at the demodulator for a 2 MHz channel bandwidth) [1], this receiver achieves a −94 ± 0.5 dBm sensitivity

$$P_{sen} = -174\,\text{dBm/Hz} + \text{NF} + 10\log B + \text{SNR}_{min}. \qquad (6)$$

Based on [42], a figure of merit (FoM) suitable for ultra-low power receivers is given by

$$\text{FoM} = \text{Sensitivity} \cdot \text{Power}^2 \cdot \text{Area}. \qquad (7)$$

In this FoM, the sensitivity indicates the large receiving distance of the receiver, the power consumption indicates the lifetime of the designed receiver, and the area indicates the ability of the designed receiver to be unobtrusive and a ubiquitous operation. Therefore, lower FoM represents better receiver performance. This proposed receiver shows the lowest FoM value.

## 4. Conclusions

An ultra-low power receiver with NPSC networks is presented. The receiver operates flexibly at the multi-ISM-bands. Using the sub-threshold-biased devices and the passive NPSC networks, the receiver is able to operate from 0.4 V supply and achieves the ultra-low power consumption of 320 μW. Moreover, the NPSC networks optimize the out-of-band rejection of the receiver. The receiver is implemented in a 90 nm CMOS technology and obtained a conversion gain of 40 ± 2 dB, a noise figure of 10 ± 0.5 dB and an OB-IIP3 of −0.3 ± 2 dBm. Due to its ultra-low power consumption, the designed receiver can be widely used in IoT devices powered by environmental energy.

**Author Contributions:** Conceptualization, L.L.; methodology, L.L.; software, F.H.; validation, Q.Q. and X.W.; data curation, L.L.; writing—original draft preparation, L.L.; writing—review and editing, Z.L. and W.W.; project administration, W.W. All authors have read and agreed to the published version of the manuscript.

**Funding:** This work was supported in part by the National Natural Science Foundation of China, grant number 61801027; 111 Project of China, grant number B14010; Beijing Nova Program of Science and Technology, grant number Z191100001119078.

**Conflicts of Interest:** The authors declare no conflict of interest.

## Abbreviations

The following abbreviations are used in this manuscript:

| | |
|---|---|
| ULP | Ultra-low power |
| NPSC | N-path switched-capacitor |
| LNA | Low noise amplifier |
| IoT | Internet of Things |
| LPTV | Linear periodically time-variant |
| FoM | Figure of merit |

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
