# Peer review of "A 320 μW Multi-Band Receiver with N-Path Switched-Capacitor Networks"

_electronics, doi:10.3390/electronics11244111_

Round 1

Reviewer 1 Report

I congratulate the authors for a job well done not only in designing the Multi-Band Receiver with N-Path Switched-Capacitor Networks but also in processing the text for the paper. I have only minor remarks:

Please consider an explanation of abbreviations when they are used for the first time, as ISM in the abstract.

Figure 5. description  „(a) Proposed LNA with an NPSC network and its equivalent RLC circuit. (b) Small-signal equivalent circuit“ does not match what is in the picture. I would guess the simulation results for four ISM bands of 430 MHz, 860 MHz, 915 MHz and 960 MHz.

Author Response

Reviewer#1, Concern # 1: Please consider an explanation of abbreviations when they are used for the first time, as ISM in the abstract.  

Author response: Thank you very much for your comments. The abbreviation of ISM in the abstract is explained.

Author action: We updated the manuscript to explain the abbreviation of ISM in the abstract.

Reviewer#1, Concern # 2: Figure 5. Description “(a) Proposed LNA with an NPSC network and its equivalent RLC circuit. (b) Small-signal equivalent circuit” does not match what is in the picture. I would guess the simulation results for four ISM bands of 430 MHz, 860 MHz, 915 MHz and 960 MHz.

Author response: Thanks for your valuable comments, we are sorry for the confusion. The description of Figure 5. is corrected as "Simulated (a) S11 and (b) Gain of the LNA for four ISM bands of 430 MHz, 860 MHz, 915 MHz and 960 MHz."

Author action: We updated the manuscript by correcting the description of Figure 5. as "Simulated (a) S11 and (b) Gain of the LNA for four ISM bands of 430 MHz, 860 MHz, 915 MHz and 960 MHz."

Reviewer 2 Report

Interesting paper for multi-band receiver with n-path switched-capacitor.  I have the following recommendations for the work:

1)       Expressions 3 and 4 are shown on page 4. It is not made clear whether they are derived by the author or taken from the literature. The variables in them and how they relate to the diagram shown above are not explained.

2)       The caption under Figure 5 is "Small-signal equivalent circuit", and this is not a circuit but results from ......

3)       On page 5, line 137 "simulation 137 results for four", the paper does not specify the model of the figure in what software it was developed, under what conditions the simulation was conducted, Which parasitic elements of the switches (transistors) to take into account. Whether ideal elements were used.

4)       On page 7, lines 194 to 198, results for power consumed by individual units are commented, it is not clear how these powers were measured.

5)       In conclusion, interesting work, but perhaps to maintain confidentiality of part of the development no details of the measurements, simulations and implementation of the converter are explained. Since the focus is on power reduction it is good to clarify how the corresponding powers were measured. I suggest that the paper be published after completion for clarification.
